# Eliciting women's preferences for place of child birth at a peri-urban setting in Nairobi, Kenya: A discrete choice experiment

**Jackline Oluoch-Aridi**[1,2]*, **Mary B. Adam**[1,3], **Francis Wafula**[1], **Gilbert K'okwaro**[1]

**1** Institute of Healthcare Management, Strathmore University, Nairobi, Kenya, **2** The Ford Family Program in Human Development Studies & Solidarity, Kellogg Institute of International Studies, University of Notre Dame, Indiana, United States, **3** Maternal Newborn Community Health, AIC Hospital, Kijabe, Kenya

* Jackline.A.Oluoch-Aridi1@nd.edu

**Data Availability Statement:** The The data underlying the results presented in the study are available from https://figshare.com/articles/Peri-urban_DCE_and_baseline_dataset/11933568

## Abstract

### Objective

Maternal and newborn mortality rates are high in peri-urban areas in cities in Kenya, yet little is known about what drives women's decisions on where to deliver. This study aimed at understanding women's preferences on place of childbirth and how sociodemographic factors shape these preferences.

### Methods

This study used a Discrete Choice Experiment (DCE) to quantify the relative importance of attributes on women's choice of place of childbirth within a peri-urban setting in Nairobi, Kenya. Participants were women aged 18–49 years, who had delivered at six health facilities. The DCE consisted of six attributes: cleanliness, availability of medical equipment and drug supplies, attitude of healthcare worker, cost of delivery services, the quality of clinical services, distance and an opt-out alternative. Each woman received eight questions. A conditional logit model established the relative strength of preferences. A mixed logit model was used to assess how women's preferences for selected attributes changed based on their sociodemographic characteristics.

### Results

411 women participated in the Discrete Choice Experiment, a response rate of 97.6% and completed 20,080 choice tasks. Health facility cleanliness was found to have the strongest association with choice of health facility ($\beta$ = 1.488 p<0.001) followed respectively by medical equipment and supplies availability ($\beta$ = 1.435 p<0.001). The opt-out alternative ($\beta$ = 1.424 p<0.001) came third. The attitude of the health care workers ($\beta$ = 1.347, p<0.001), quality of clinical services ($\beta$ = 0.385, p<0.001), distance ($\beta$ = 0.339, p<0.001) and cost ($\beta$ = 0.0002 p<0.001) were ranked 4th to 7th respectively. Women who were younger and were the main income earners having a stronger preference for clean health facilities. Older married women had stronger preference for availability of medical equipment and kind healthcare workers.

https://figshare.com/articles/Appendix_8_Mother_
Data_deidentified_JoluochARIDI_2020_xlsx/
11925858 https://figshare.com/articles/Do-file_
for_peri-urban_DCE_in_Kenya/11933610 https://
figshare.com/articles/Stata_Analysis_dta_file_for_
peri-urban_paper_on_women_s_preferences_for_
place_of_delivery_in_a_peri-urban_setting_Kenya/
11926272.

**Funding:** This study was funded by the Kellogg
Institute of International studies by a grant to the
corresponding author. The funders had no role in
study design, data collection and analysis, decision
to publish, or preparation of the manuscript.

**Competing interests:** The authors have declared
that no competing interests exist.

## Conclusions

Women preferred both technical and process indicators of quality of care. DCE's can lead to the development of person-centered strategies that take into account the preferences of women to improve maternal and newborn health outcomes.

## Introduction

In 2015, an estimated 303,000 women were reported to have died in the developing world out of maternal causes [1]. Nearly all maternal deaths (99%) occur in developing countries with over half of these deaths occurring in sub-Saharan Africa [1]. This is a key reason behind Sustainable Development Goal 3 (SDG3), which aims to reduce the global maternal mortality ratio (MMR) to 70 for every 100,000 live births by 2030 [2]. Skilled attendant delivery is considered an effective way of achieving this yet evidence suggests that it remains low, particularly across rural and poorer urban locations [3, 4]. The UN projects that by 2020, most Africans will be living in urban areas with rapid urbanization resulting in informal settlements and posing additional challenges to access to maternal health services [5].

Kenya's maternal mortality ratio of 362 per 100,000 live births is relatively high for a lower middle income country [6]. The ratios are particularly high across peri-urban informal settings. One study reported an maternal mortality ratio of 700 per 100,000 livebirths in two informal settlements in Nairobi [7]. The high ratios have been attributed to inequities across peri-urban settings that residents within cities face [5]. To improve maternal health outcomes, the Kenyan Government made delivery services free at public health facilities in 2013 [8]. The free delivery policy substantially reduced out-of-pocket costs leaving families to cover incidental charges only. Health centers and dispensaries received direct reimbursements through the hospital Sector Services Fund to an amount of (2500 Ksh /25$). Hospitals that offered referral care including C-sections were directly reimbursed through the Hospital Management Service Fund (Ksh 5000/$50) [9]. Women continued to pay incidental charges which varied by facility and presumably were for supplies that were not part of the hospital reimbursement package from the government.

Early evidence suggests that the policy may have increased facility delivery up from 86% to 95% in five urban settings in Kenya [10]. Another study reported an increase of 27% and 16% for deliveries and antenatal services across county referral hospitals and low-cost private facilities respectively [11]. On the other hand, studies showed that health systems preparedness was inadequate, posing danger of poor quality [12]. Challenges reported included delays in health facility reimbursements for deliveries, lack of ambulances for referral, stock outs of essential supplies and health worker shortages [13–15]. For that reason, there is increased awareness on the need to focus on both access and quality of service [14]. Quality is measured on the basis of safety, effectiveness, timeliness, efficiency, equity and patient-centeredness [15]. The Lancet Commission on High Quality Health Systems estimated that one in three people in low and middle income countries (LMIC's) have had a negative experience with the health system in the areas of attention, respect, communication and length of visits and disrespectful treatment [16]. Evidence also indicates that poorer women, such as those in the peri-urban settings, have a higher likelihood of encountering poor quality of maternal health services in Kenya [17]. While there has been increased focus on quality, [12, 13] efforts have mainly focused on health system inputs and satisfaction at the end of the continuum of care- both of which fail to identify and rank the relative weight of demand side barriers to access [18]. Such studies cannot fully explain what drives women to choose a facility. This information is particularly useful for prioritization in resource constrained settings. This is where Discrete Choice Experiments (DCE) have added value.

DCE's allow health services users to state individual preferences based on predefined hypothetical choices. They are based on the assumption that services can be described by their attributes, and that the value of a service depends on the nature and level of these attributes [19]. The theoretical basis for Discrete Choice Experiments (DCEs) is described elsewhere [19, 20]. DCE's have been used to examine a broad range of health system challenges in sub-Saharan Africa [21], patient preferences for hospital services [22] and maternal health services in rural areas of Tanzania and Ethiopia [23–25]. The studies have mainly focused on women in rural settings. The few studies that have looked at urban settings have focused on other determinants of delivery [26–29]. This study uses the DCE methodology to understand the attributes of the health system that women value most when making the decision on where to deliver.

## Materials and methods

### Study setting

The study was conducted at Embakasi-North, a sub-county in Nairobi County with a population of 181,388 people and is located about 10 km to the East of Nairobi City. Embakasi-North is home to Dandora, an area that houses the largest municipal dumpsite in Nairobi, and is characterized by low-income residential housing estates. The area is served by a mix of public, private and faith-based facilities of different levels. Mama Lucy Maternity Hospital, a secondary referral hospital, is located in the neighboring sub-county. Maternity facilities utilized by women in these informal settlements vary widely in terms of quality of care that they provide. The facility-based delivery rate in Nairobi is high with approximately 88.7% of women delivering within a health facility [6]. However, within peri-urban settings and informal settlements in Nairobi have been known to have lower rates of facility-based delivery [6].

### Development of DCE attributes and attribute levels

The study entailed conducting a literature review and doing a qualitative study to determine attributes and attribute levels that were important to women. The qualitative study sought to explore the perceptions and experiences of women visiting health facilities in the area. The results of the qualitative study can be found here [30]. After obtaining informed consent from the women, trained facilitators led the focus group discussions (FGDs). Women were asked to explain how they made the choices and identify which facility features drove their child birth choices. Women were purposively selected and each FGD had 6–8 women. The characteristics of the 40 women interviewed are contained in (S1 Appendix).

Qualitative Interview data were entered into *Nvivo 11* and coding done. Thematic analysis was done following the six key steps, namely, familiarization with the data, coding, grouping codes, identifying themes, additional coding and refining of themes, and writing up the results. Four broad themes were identified: perceived quality of delivery services, financial access, physical amenities at the facility, and health worker's strike. (See S2 Appendix). The themes helped in deriving attributes and attribute levels. The selected attributes were piloted on 30 women residing outside of the study setting in a neighboring sub-county to test for suitability and the cognitive response of the women in understanding the selected attributes.

The pilot showed that the attributes could be easily understood and traded-off by the women. Some attributes such as the costs of delivery attributes were revised and were chosen based on what the women reported they had paid when going to deliver. The costs ranged from 3000 to 8000 Ksh for normal uncomplicated deliveries in both the public and private health facilities. The costs were inclusive of out-of-pocket costs that the women were charged during delivery. These costs were present even at facilities that had the "free delivery" policy. For a complete list of the attributes and attribute levels selected for the DCE, See Table 1.

**Table 1. List of attributes and attribute levels included for the DCE.**

| Attribute | Attribute level |
|---|---|
| Quality of clinical services at the health facility | Good quality of clinical services |
| | Bad quality of clinical services |
| Attitude of healthcare workers | Kind and supportive healthcare worker |
| | Unkind and unsupportive healthcare worker |
| Availability of medical equipment and supplies | Medical equipment and supplies available |
| | Medical equipment and supplies not available |
| Distance to the health facility | Health facility is close to residence |
| | Health facility is far from residence |
| Cleanliness of the health facility | Clean health facility |
| | Dirty health facility |
| Cost of delivery service | 3000; 5000; 8000 |

*Note. Costs are in Ksh (1 USD = 100Ksh) Costs are not zero even with free delivery policy due to incidental fees charges at government facilities.

## The DCE experimental design

The study was designed as an unlabeled DCE with sixteen choice set presented under three alternatives: alternative of health facility A, alternative of health facility B, and an opt-out alternative where the woman would choose none of the two facilities, explained as a preference for home delivery. S3 Appendix shows a sample choice-card with a scenario showing the final attributes and attribute levels included. The attributes of the health facility were explained to the women using a choice-card that contained a brief description of the definition of the attributes. For example. Cleanliness meant a health facility that had a clean ward with clean beds, bathrooms and toilets (See S4 Appendix).

All attributes in the choice experiment were dichotomous, except cost, which had three levels. This resulted in a design of $(2^5 \times 1^3) = 96$. The number of alternatives of attribute levels in the full fractional design was calculated to $(96*95)/2 = 4560$. A fractional factorial design helped to reduce the choice-sets to 16, making it simpler for the respondents. We used JMP software for a D-efficient experimental design and resulted in a D-error of 0.3 (*JMP Pro*). (See S5 Appendix). The D-efficient design also allowed for favorable design such as orthogonality, level balance, minimum balance and overlap [31].The 16 choice-set questions were generated from the design. The choice-sets were grouped into two through a process called blocking using ODK software and each woman answered eight questions in a single block.

## Data collection for the household and the DCE survey

Following administration of informed consent, a random sample of women of reproductive age (18–49 years) were recruited from a larger household survey in the area. The inclusion criteria were women who had delivered in the past five years. The main household questionnaire was a composite tool carrying questions from the Kenya Demographic Health Survey and the African Population and Health Research Survey [5, 6]. The survey contained questions on women's sociodemographic characteristics and maternal health services utilization variables. For The questionnaire (See S6 Appendix), and details of the sampling process from the larger household survey provided in (See S7 Appendix). The sample size for the DCE was calculated using the Johnson and Orme methodology [32]. The household survey was conducted between August and September 2017 by trained research assistants using Open Data Kit (ODK) platform. This was followed by the DCE survey, which asked women to imagine a hypothetical scenario where they were expecting a baby and had

to choose between facilities A and B for delivery (or none). The women were told that the opt-out option implied home delivery. They were also told that there were no wrong or right answers, and that they were free to stop the experiment at any time (See S8 Appendix).

## Ethics approval

Ethics approval for the study was provided by the African Medical Research Foundation (AMREF) research Committee, the National Commission for Science and Technology (NACOSTI) as well as the Country Directors of health in charge of the sub-county.

## Data analysis

The DCE data was analyzed using the random utility model, a model that expresses the utility 'U' in of an alternative $i$ in a choice set $C_n$ (perceived by individual $n$) as two parts: 1) An explainable component specified as a function of the attributes of the alternatives $V(X_{in}, \beta)$; and 2) an unexplainable component (random variation) $\varepsilon_{in}$. [33].

$$U_{in} = V(X_{in}, \beta) + \varepsilon_{in}$$

The individual $n$ will choose alternative $i$ over other alternatives in a choice set C if and only if this alternative gives the maximized utility. The relationship between the utility function and the observed $k$ attributes of the alternatives can be assumed under a linear-in-parameter function [34]. Therefore, the utility the respondents attach is related to the attribute and attribute levels within the choice-sets, meaning that if alternative i is chosen within a choice set, i will yield the maximum utility compared to j alternatives. A is the alternative specific constant, x are the attributes in the DCE and β are the coefficients describing the marginal utility of the attribute. The standard conditional logit model is below:

$$V_{in} = \alpha_i + \beta_i x_{i1} + \ldots + \beta_k x_{i+e}$$

The data were imported and analyzed in Stata 15 (*StataCorp LP, College Station, USA*). Descriptive statistics were calculated for the non-DCE variables. The cost attribute was assumed to be linear while all other attributes were categorical variables, therefore non-linear. A base conditional model was used to estimate the mean change in utility, preference which respondent placed on attributes [34].

$\alpha i$ is a constant term that represents the general preference for place of delivery at a health facility compared to the alternative of opting out and having a home delivery. Dummy coding was used for the data, each attribute level was assigned a value of 1 whenever it was retained and 0 when omitted. The cost of delivery service was entered in the model as a continuous variable. All the other five variables were coded as categorical variables. The Utility Model makes the assumption that women will trade-off between the different attribute levels and choose the alternative that gives the greatest utility. The conditional model is suitable for estimating average preferences across respondents. The utility function was estimated for the following model:

$U_i = \alpha_i + \beta1GoodQualClin + \beta2BadQualClin + \beta3kindattitudeofhealthworkers + \beta4unkind$

$andunsupportiveattitude + \beta5Medequipavail + \beta6Medequipnotavai + \beta7Shortdist + \beta8longdist +$

$\beta9cleanclean + \beta10cleandirty + \beta11Costs + \varepsilon$ (error term)

$\alpha_i$ is the alternative specific constant (ASC) term that shows the preference for place of delivery (either a health facility or home), β's 1–11 are the parameters for each of the attribute levels and $\varepsilon$ is the error term.

The dependent variable is the place of delivery represented by the unlabeled choices health facility A, health facility B and the opt-out (home delivery), while the independent variables are the respective attribute levels of the characteristics of the place of delivery. The base conditional logit model assumed homogeneous preferences across respondents [34]. The output of the conditional logit model contains the beta which shows the magnitude of the preferences for the attribute. Due to the assumption of irrelevant independent alternatives, the presence of heterogeneity in choices we estimated a generalized mixed logit model to assess for preference heterogeneity amongst the women [35].This was done by extending the generalized model and testing interactions between the sociodemographic and the women's attributes in order to investigate how preferences may vary according to observed individual characteristics. The sociodemographic characteristics that were included as interaction terms include sociodemographic characteristics that have been known to influence place of delivery in Kenya were also included such as maternal age, marital status, education and income status [36–39].

The output of the mixed logit model includes both the mean and the standard deviations of the random parameter estimates with confidence levels. The mean parameter estimate represents the relative utility of each attribute while the standard deviations for a random parameter suggest the existence of heterogeneity in the parameter estimates over the sampled population around the mean parameter estimate i.e., different individuals possess individual-specific parameter estimates that may be different from the sample population mean parameter estimates [35]. The p-value of the interactions shows statistical significance for an interaction between sociodemographic variables and attributes hence signifying the influence of the woman's characteristics. Insignificant parameter estimates for derived standard deviations indicate that the dispersion around the mean is statistically equal to zero, suggesting that all information in the distribution is captured within the mean. The theoretical validity of the design will be explored by examining the signs and significance of parameter estimates [35]. A correlation matrix analysis was also done to ensure that there is no inter attribute correlations between certain attributes that are close in semantic meaning (See S9 Appendix).

## Results

### Participants' characteristics

A total of 481 women were selected for the interview. Of the women, 85% were married, 53% had secondary education and 58% had at least one child. Only 11% of the women identified themselves as main earners. The participants' characteristics are summarized in Table 2. A total of 421 (87.5%) did the DCE. Ten women did not complete the DCE exercise, reducing the sample for analysis to 411 women. The total number of observations analyzed for the DCE was 20,080.

The majority of the participants (74%) had intended pregnancies and had planned to deliver at a health facility. More than half of the participants (58%) reported attending the recommended four antenatal clinic visits. Nearly 79% had no health insurance cover. Only 5% said they were referred to a tertiary health facility for delivery. The health system utilization variable are summarized in Table 3. Additional analyses on place of delivery can be found at (S10 Appendix).

### The Conditional model results

The Conditional model results (Table 4) indicate that all attributes for place of delivery were statistically significant, meaning that they were all valued by the women. The variables with the strongest association were cleanliness, availability of medical equipment and drugs, and the opt-out alternative (home delivery) in order of strength. Health worker attitude, quality of

**Table 2. Sociodemographic characteristics of the women in a peri-urban setting who were administered for DCE survey (N = 411).**

| Sociodemographic variables | N | (%) |
|---|---|---|
| Age n (mean (SD)) | 24 (0.2) | |
| Marital status | | |
| Single | 63 | 15 |
| Married | 348 | 85 |
| Education | | |
| Primary School | 14 | 34 |
| Secondary School | 220 | 53 |
| University/tertiary | 48 | 13 |
| Parity | | |
| 1 | 240 | 58 |
| $\geq 2$ | 171 | 42 |
| Is the main earner | | |
| Woman not main earner | 366 | 89 |
| Woman is main earner | 44 | 11 |
| Head of Household education | | |
| Primary school | 67 | 16 |
| Secondary school | 187 | 45 |
| University/Some tertiary | 157 | 39 |
| Woman's influence on decision making within the household | | |
| Woman had no influence in decision making | 43 | 11 |
| Woman had influence in decision making | 368 | 89 |

clinical services provided, distance and cost were fourth, fifth, sixth and seventh respectively. One finding that was rather unexpected was a positive coefficient for the cost variable, which would suggest that the women had a utility for high delivery costs. See (S11 Appendix) for more details on the conditional and mixed multinomial logit stata output.

## The generalized mixed logit model

For the generalized mixed multinomial logit model with no interactions, all the mean coefficients values for all the attributes, including the opt-out, were statistically significant at the 5% level (Table 5). This meant that we could reject the null hypothesis that stated that the attributes selected were not important to the women respondents. However, the opt-out option had a lower significance level, (at the 10% level) implying that there was less variance in the characteristics of the respondents who chose this option. All the attributes had statistically significant parameter estimates for the standard deviation, except the attributes on cost and distance. This implied that there was insufficient variation for individual specific parameter estimates that might be different from the sample population mean meaning all the information for the attribute of cost and distance was contained in the mean parameter.

## Mixed logit model with interactions between sociodemographic variables and attributes

Additional analyses sought to examine the interaction between preferred attributes and selected sociodemographic variables shown in previous studies to have an association with health facility use. These variables included age, secondary education, marital status and

**Table 3. Health system utilization variables for women in peri-urban setting women (N = 411).**

| Health system utilization variables | N | (%) |
|---|---|---|
| Pregnancy intentions | | |
| Pregnancy not intended | 106 | 26 |
| Pregnancy intended | 305 | 74 |
| Place of delivery | | |
| Public health facility | 235 | 57 |
| Private health facility | 176 | 42 |
| Place of delivery (level of delivery facility) | | |
| Home | 27 | 6 |
| Tertiary | 318 | 75 |
| Primary | 65 | 16 |
| Planning for delivery | | |
| Not planned health facility | 100 | 24 |
| Planned health facility | 311 | 76 |
| Number of ANC visits attended | | |
| ANC visits <4 | 171 | 42 |
| ANC visits >4 | 240 | 58 |
| Referral status | | |
| Not referred to a tertiary health facility | 390 | 95 |
| Referred to tertiary health facility | 20 | 5 |
| Specialist services | | |
| Did not see specialist | 356 | 87 |
| Saw a specialist | 55 | 13 |
| Whether the woman had a cesarean section | | |
| Didn't deliver via cesarean | 356 | 87 |
| Had a cesarean section | 55 | 13 |
| Health Insurance status | | |
| Didn't have health insurance | 326 | 79 |
| Had health insurance | 85 | 21 |
| When the woman moved to the peri-urban setting | | |
| Moved within the last 5 years | 219 | 53 |
| Moved to area over 5 years ago | 192 | 47 |

main earner status. The results of the interactions are in Table 6. All interactions between the attributes of cleanliness and all the interaction covariates were significant with the exception of marital status. The availability of medical equipment and drug supplies had statistically significant mean parameter estimates, with all covariates with the exception of marital status and main earner status. All the interactions for the interactions between the attitude of healthcare workers and the covariates were statistically significant at the 95% level. The only significant interaction was between quality of clinical care services with secondary education.

Younger unmarried women with a secondary education and who self-identified as main income earners in their household had a significant strong preference for clean health facilities. Older women with a secondary education showed a strong preference for a health facility with availability of medical equipment and supplies. All the interaction between the attitude of healthcare workers and age, marital status and main earner status had a statistically significant parameter estimate. Older women, married women with a secondary education and who identified themselves as main earners all showed a strong

**Table 4. Conditional Logit model for a discrete choice experiment assessing women's preferences in a peri-urban setting in Kenya (N = 411).**

| Attribute | β | S. E | P value | C.I | Expected sign |
|---|---|---|---|---|---|
| Clean (Cleanliness) Ref | 1.488 | 0.434 | <0.001 | (1.403–1.573) | + |
| Dirty | | | | | |
| Medequip (Available) Ref | 1.435 | 0.046 | <0.001 | (1.343–1.527) | + |
| Medequip (Unavailable) | | | | | |
| ASC (opt-out) | 1.424 | 0.134 | <0.001 | (1.162–1.685) | + |
| Attitude (Kind and supportive) Ref | 1.347 | 0.038 | <0.001 | (1.272–1.423) | + |
| Attitude (Unkind and unsupportive) | | | | | |
| Qualclin (Good) Ref | 0.385 | 0.045 | <0.001 | (0.297–0.473) | + |
| Bad | | | | | |
| Distance (Short) Ref | 0.339 | 0.037 | <0.001 | (0.267–0.412) | + |
| Distance (Long) | | | | | |
| Cost | 0.0002 | 0.0000135 | <0.001 | (0.0002–0.00024) | - |
| No. of Observ. | 20208 | | | | |
| Pseudo R$^2$ | 0.5410 | | | | |
| Wald Chi | 4564.83 | | | | |
| Prob >chi2 | 0.0000 | | | | |
| Log likelihood | -3396.87 | | | | |

(Clean- Cleanliness of the health facility, Medequip- Medical equipment and drugs, ASC-Alternative Specific Constant, Attitude- Attitude of healthcare workers, Qualclin.-Quality of the clinical delivery services, Distance- Distance to the health facility) REF- reference category Observ.- Observations

preference for a kind and supportive healthcare worker. Only women with secondary schooling showed a strong preference for quality of clinical care. Lastly all the interactions between the variables of cost and distance were statistically insignificant at the 95% level. See Tables 5 and 6 below for the details on the base model (generalized mixed logit and the interactions model with the selected sociodemographic variables respectively).

## Model estimates for preference for attribute levels

The results showed that both models (conditional logit and mixed logit) were statistically significant (p < 0.05). The chi squared tests were also significant for both models (p<0.0001). The log pseudo likelihood of the conditional multinomial logit was -3396.87 and the generalized mixed conditional model was -2813.69. The models with interactions provided improved explanatory power for the generalized mixed logit model. See (S12 Appendix) for details on the Log Likelihood Ratio Test.

**Table 5. Mixed logit model results with and without interactions for a discrete choice experiment addressing facility preferences for delivery among women, in a peri-urban setting in Kenya (N = 411).**

| Attribute | Base model | | Interaction terms (Mean Parameter) w/ Sec educ | | w/ age | | w/ marital status | | w/main earner | |
|---|---|---|---|---|---|---|---|---|---|---|
| | β[a] | P-value | β[a] | P-value | β[a] | P-value | β[a] | P-value | β[a] | P-value |
| Medequip (Available) Ref | 2.266* | <0.001 | 1.660* | <0.001 | 0.047* | <0.001 | 0.051 | 0.606 | 0.871 | <0.001 |
| Cleanliness (Clean) Ref | 2.258* | <0.001 | 1.769* | <0.001 | -0.037* | 0.007 | -0.282* | <0.001 | 1.649 | <0.001 |
| Attitude (Kind & supportive) Ref | 2.039* | <0.001 | 1.681* | <0.001 | 0.124* | 0.001 | 1.155* | 0.001 | 0.246 | 0.470 |
| QualClin (Good) Ref | 0.570* | <0.001 | 0.575* | <0.001 | -0.017* | <0.001 | -0.036 | 0.614 | -0.090 | 0.285 |
| Distance (Close) Ref | 0.445* | <0.001 | 0.467* | <0.001 | 0.017* | <0.001 | 0.439* | <0.001 | 0.399 | <0.001 |
| Cost, Ksh[b] | -8.091* | <0.001 | -8.166* | <0.001 | -11.271 | <0.001 | -8.102 | <0.001 | -8.262 | <0.001 |

**Table 6. Mixed logit model results with interactions for a discrete choice experiment addressing facility preferences for delivery among women, in a peri-urban setting in Kenya.**

| | | Interaction terms (SDs) | | | | | | | |
| --- | --- | --- | --- | --- | --- | --- | --- | --- | --- |
| | | w/seco educ. | | w/age | | w/marital status | | w/main earner | |
| | | β | p-value | β | p-value | β | p-value | β | p-value |
| Medequip X covariate | | 0.330*** | <0.001 | 0.007*** | 0.001 | 0.042 | 0.333 | 0.093 | 0.456 |
| Clean X covariate | | 0.389*** | 0.001 | 0.014*** | 0.016 | 0.164 | 0.456 | 0.562*** | 0.001 |
| Attitude covariate | | 1.314*** | <0.001 | 0.037*** | 0.002 | 0.611*** | 0.001 | 1.106*** | 0.001 |
| QualClin X covariate | | 0.698*** | <0.001 | 0.0014 | 0.228 | -0.004 | 0.491 | 0.009 | 0.727 |
| Distance X covariate | | 0.008 | 0.700 | -0.0001 | 0.860 | -0.0001 | 0.993 | -0.011 | 0.778 |
| Cost X covariate | | 0.043 | 0.208 | 0.0013 | 0.986*** | -0.031 | 0.378 | 0.07 | 0.679 |
| **Respondents** | | 411 | | 411 | | 411 | | 411 | |
| **Log Likelihood** | -2831.7 | | | | | | | | |
| **Prob> χ2** | 0.0000 | | | | | | | | |

## Validity of the data

The theoretical validity was also checked through comparing the expected direction of the parameter estimates and this was found consistent with expectations with the exception of the attribute on costs, which had a positive coefficient. One would generally expect it to have a negative one because this signifies a utility for lower costs for delivery services. It did have a negative coefficient in the mixed logit analyses.

## Discussion

To the best of our knowledge, this study is the first DCE study conducted in a peri-urban setting in sub-Saharan Africa. Previous studies in sub-Saharan Africa such as those done in and Zambia, Tanzania, and Ethiopia were conducted in rural settings [22–25]. The study found that all the attributes had an effect on the decision on where to deliver. The women in this setting highly valued aspects of quality related to the technical quality such as the cleanliness of the health facility and the availability of equipment and supplies. They also valued aspects related to processes involved in delivery, particularly, the attitude of healthcare workers and to a lesser extent clinical quality during delivery.

The most valued attribute was health facility cleanliness, suggesting that relatively less costly interventions such as facility hygiene may increase skilled attendant deliveries in such settings. Cleanliness has also been identified as an important attribute in Ethiopia and in Zambia [22, 23]. Studies using other methods done in maternity settings in urban Kenya have identified similar factors associated with satisfaction, including waiting time, attitude of the health providers, availability of drugs, affordability of services, staffing level and cleanliness [40]. Yet, other studies have shown hygiene to be poor across facilities in informal settlements [27]. The Kenya Quality Model for Health (KQMH), the Kenyan government's quality management framework, emphasizes hygiene as a primary intervention across facilities [41] Poor hygiene may cause users to avoid a facility and travel longer distances for care, which can cause delays and worsen maternal indicators. Such behavior has been documented in Tanzania for instance [42].

The second most highly ranked attribute was availability of medical equipment and drug supplies. This was also reported in Ethiopia [23, 24]. A recent study in Kenya found that only two of five health centers assessed had acceptable emergency obstetric care capability [43]. Additionally recent assessments at Kenyan health facilities found that essential medical equipment and drug supplies were unavailable at 31% at public health facilities 59% of private health

facilities [18]. However evidence shows that availability of infrastructure such as equipment may not necessarily translate into effective coverage for obstetric complications [44].

The kind and supportive attitude of healthcare workers was ranked third. Similar findings have been reported across Africa, including Zambia, Tanzania and Ethiopia [22–24], A recent review reported that the attitude of health workers managing women during labor and delivery presented a major quality challenge across low income settings [45]. Similar findings have been reported locally [46–48], and other African countries [49]. As a result, there has been more emphasis on promoting accountability for the actions of health care workers with regard to mistreatment of women during delivery [50, 51].

When attributes were interacted with sociodemographic characteristics that are known to influence women's preferences for health facility for delivery, we found that younger unmarried women with a secondary education who were main earners had a stronger preference for clean health facilities. As expected women with a secondary education also had a strong preference for health facilities with medical equipment and supplies, health workers with a kind attitude and good quality care by health care workers The preference is consistent with local literature and other low-income countries with secondary educated women preferring to delivery in health facilities that signal high quality [38, 52]. Peri-urban settings are increasingly populated with women with secondary education and it is important for the health system to be responsive to their preferences [5].

Older women on the other hand expressed the strongest preference for health facilities with medical equipment and supplies. Reasons are not clear. It may be that older women have more experience with the health system and have higher expectations. This then drove them to choose health facilities that had these types of equipment. Older women might also strongly rely on social networks for decision making on place of delivery [53].

The attributes on cost and distance did not have significant preference heterogeneity. This implies that there was no variation in the characteristics of individual women who chose these attributes. Attributes such as distance that have been reported as important attributes in certain contexts appeared to have lower value across the study setting, possibly because the women could geographically access several healthcare facilities easily. A rural setting DCE study reported distance to be lowly ranked attribute [23]. The low effect of distance may partly explain why attributes that signal quality such as cleanliness were the most important [23]. Policy may build on this observation to push for quality improvement at healthcare facilities, although the concern is that less visible aspects of quality may be ignored.

## Strengths and limitations of the study

The main limitation of the study was the challenge in developing certain attributes accurately. This might have led to the underestimation of the parameter estimates such as the cost leading to a positive coefficient instead of a negative. Also, difficulty in definition of the attribute on quality of clinical services led to a relatively weak specification, mainly because the translation of the word 'treatment' carries both clinical and non-clinical (client relation) connotation to the women. The women may not have been able to separate the two. The gold-standard for assessing clinical quality is measuring staff adherence to guidelines, something that was beyond the scope of this study. However, effort was made to improve the theoretical validity of the study through doing the qualitative study to help identify new attributes of importance in the context, or confirm the ones identified in literature.

## Conclusion

In conclusion, understanding the relative contribution of factors that influence the choice of a place for delivery is important for policy makers who are aimed at reducing maternal and

newborn deaths as well as improving the quality of care. In exploring the complex context of facility-based delivery for women it is important for policy makers to have a deeper understanding on women's preferences this will help in overcoming barriers to high quality delivery care and structuring more patient-centered services that can lead to improved maternal health outcomes.

## Supporting information

**S1 Appendix. Characteristics of women interviewed in focus group discussion.**
(DOCX)

**S2 Appendix. Qualitative paper.**
(PDF)

**S3 Appendix. Example of a scenario in a choice-set card that was presented to the women.**
(DOCX)

**S4 Appendix. DCE sample choice card information packet.**
(PDF)

**S5 Appendix. DCE experimental design.**
(PDF)

**S6 Appendix. The household survey questionnaire.**
(PDF)

**S7 Appendix. Sampling points for the household survey.**
(DOCX)

**S8 Appendix. Digital presentation of the DCE experiment on open data kit for women.**
(DOCX)

**S9 Appendix. Correlation matrix.**
(DOCX)

**S10 Appendix. Stata output on place of birth variable.**
(DOCX)

**S11 Appendix. Conditional and mixed multinomial logit status output.**
(PDF)

**S12 Appendix. Log likelihood ratio test.**
(DOCX)

**S13 Appendix. Mother data de-identified.**
(XLSX)

## Acknowledgments

The authors are first of all grateful to the Embakasi-North health management team, the women and healthcare workers in Embakasi-North for allowing us to interview them. We would like to thank our research assistants; Cindy Mical, Brian Ambutsi, Christine Achieng and Mercy Ngao. We appreciate Melvin Obadha, Maurice Baraza and Dr. Sydney Oluoch for assisting with the data analysis. We thank the Institute for healthcare Management PhD seminar group; Dr. Ben Ngoye, Dr. Tecla Kivuli and Eric Tama for their critical feedback during PhD seminars.

## Author Contributions

**Conceptualization:** Jackline Oluoch-Aridi, Mary B. Adam, Francis Wafula, Gilbert K'okwaro.

**Data curation:** Jackline Oluoch-Aridi.

**Formal analysis:** Jackline Oluoch-Aridi.

**Funding acquisition:** Jackline Oluoch-Aridi.

**Investigation:** Jackline Oluoch-Aridi, Mary B. Adam.

**Methodology:** Jackline Oluoch-Aridi.

**Project administration:** Jackline Oluoch-Aridi.

**Resources:** Jackline Oluoch-Aridi.

**Software:** Mary B. Adam.

**Supervision:** Jackline Oluoch-Aridi, Francis Wafula, Gilbert K'okwaro.

**Validation:** Jackline Oluoch-Aridi, Mary B. Adam, Gilbert K'okwaro.

**Writing – original draft:** Jackline Oluoch-Aridi.

**Writing – review & editing:** Jackline Oluoch-Aridi, Mary B. Adam, Francis Wafula, Gilbert K'okwaro.

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
