## [Decision Letter · Decision Letter 0]

20 Jan 2020

PONE-D-19-31801

Eliciting women’s preferences for place of delivery in a peri-urban setting in Kenya: A discrete choice experiment

PLOS ONE

Dear Ms Aridi,

Thank you for submitting your manuscript to PLOS ONE. After careful consideration, we feel that it has merit but does not fully meet PLOS ONE’s publication criteria as it currently stands. Therefore, we invite you to submit a revised version of the manuscript that addresses the points raised during the review process.

We would appreciate receiving your revised manuscript by Mar 05 2020 11:59PM. To enhance the reproducibility of your results, we recommend that if applicable you deposit your laboratory protocols in protocols.io, where a protocol can be assigned its own identifier (DOI) such that it can be cited independently in the future. For instructions see: http://journals.plos.org/plosone/s/submission-guidelines#loc-laboratory-protocols

We look forward to receiving your revised manuscript.

Kind regards,

Jan Ostermann

Academic Editor

PLOS ONE

Journal Requirements:

3. Your ethics statement must appear in the Methods section of your manuscript. If your ethics statement is written in any section besides the Methods, please move it to the Methods section and delete it from any other section. Please also ensure that your ethics statement is included in your manuscript, as the ethics section of your online submission will not be published alongside your manuscript.

4. We note you have included a table to which you do not refer in the text of your manuscript. Please ensure that you refer to Table 7 in your text; if accepted, production will need this reference to link the reader to the Table.

Reviewers' comments:

Reviewer's Responses to Questions

**Comments to the Author**

1. Is the manuscript technically sound, and do the data support the conclusions?

Reviewer #1: Partly

2. Has the statistical analysis been performed appropriately and rigorously? 

Reviewer #1: No

3. Have the authors made all data underlying the findings in their manuscript fully available?

Reviewer #1: Yes

4. Is the manuscript presented in an intelligible fashion and written in standard English?

Reviewer #1: Yes

5. Review Comments to the Author

Reviewer #1: Thank you very much for the chance to review the paper by Ms. Aridi et al. on eliciting women’s preferences for a place of delivery in Kenya using a discrete choice experiment. Overall, the paper reports women’s preference and relative importance for a few key attributes and costs that women may consider when deciding a delivery place in a peri-urban setting in Kenya. The author presented interesting study design, analysis and results from a reasonably well-represented study population to address the authors’ research question. However, the paper has some methodological weaknesses, incomplete and unclear presentation of the data and analytical techniques, and questionable interpretation of the results, which significantly weaken the quality of the manuscript. There are a number of places where the sentences were poorly written, and lack of line numbers also made difficult to make comments.

Major comments:

Study Design:

- Although the authors indicated that they selected the attributes and attribute levels were selected based on a qualitative study, there are no description how a qualitative study was done. Please provide detailed information about the qualitative studies (the type of the qualitative study, how many interviews were done, brief sociodemographic characteristics of these women, and how these data were analysed).

- I strongly think the two attributes, “Interpersonal treatment at the health facility’ and “Attitude of healthcare workers” are correlated. How do these two attributes differ? Furthermore, I worry that the dichotomy of “good” vs” bad” interpersonal treatment seems too simplistic, and can be unclear and very subjective to respondents. Also, I suspect that’s one of the reasons why the authors see a much lower coefficient for the interpersonal treatment.

- I think the biggest question for the DCE design is the attribute level of costs. Given that the authors say that the there are no delivery fees at public health facilities, why didn’t the authors include no cost as the level option? Are there any other associated fees women need to pay when they deliver at public health facilities? The authors did not provide a data on how many of these women had previously delivered at public health facilities, compared to private health facilities. If most of women deliver at public health facilities, I worry that the hypothetical scenarios of paying the extra money might have not been correctly judged by the respondents as designed. Also, I wonder how the authors chose the cut-offs for the costs (3000, 5000, 8000)? Please help the readers understand whether these are reasonable range of costs to deliver at the facilities (especially at the public health facilities).

- The authors mentioned that they chose a random sample out of a larger household survey. Which household survey is this (please cite it and provide a brief description)? Also, please provide more description of how this random selection was done – was there any stratification by geographic region or any other characteristics?

- Was the survey used one fixed same questionnaire for everyone? Please specify.

- Please give a detailed assumption and calculation for how the sample size was calculated using the de Dekker-Grob’s formula. What coefficients did the authors assume initially? Did the authors calculate based on the conditional logit model or mixed logit model for their initial sample size calculation? At what percent of power? What was the minimum sample size required for the study?

- Can the authors provide more details about the D-efficient design? Were orthogonality and level of balance achieved? How many did each attribute level appear in the questionnaire? What was the goodness-of-fit for the test?

- I am not sure why the authors would have blocked the questions into the two groups and ask each women answer two blocks anyway (it deficits the purpose of blocking…).

- The authors indicated conducting a pilot study. Can the authors provide more details how the results from the pilot study testing were used to improve the study questionnaire?

- The formula for possible choices is incorrect: it should be “2^5 x 3^1 = 96”

Study analysis/Results:

- The authors described that “opt-out” option was included in the model. How many people chose “opt-out” option out of all available choice tasks? Also, with about 26% of women planning not to deliver at health facilities, I worry whether there are other factors that influence women’s choices to deliver at outside of health facilities (i.e. home). Did the authors look at preference coefficients among women who chose “opt-out” option at least once? Was there anyone who mostly chose the “opt-out” option?

- Can the authors clarify the difference between Table 6 and 7?

- Table 6: I disagree with the approach that the authors put all possible combinations of different interactions between sociodemographic variables and attributes. Even so, I encourage not to show all the non-significant (especially if not driven by specific hypotheses) results in Table 6. What is log worth? The authors present acronym of the attributes which were not previously defined in the manuscript thus it is difficult to understand Table 6. Please present meaningful acronyms (then define them in the footnote) or full description of the attributes. P-value should be presented up to 2-3 digits in the table and throughout the paper.

- I am surprised that the cost coefficient in the conditional logit model is positive. More confusingly, while the cost coefficient is positive in the conditional logit (Table 4), the cost coefficient in the mixed logit model is very significantly negative. Why is that?

- Tale 6: It was not clear whether all interaction terms were fitted into one big model or whether each interaction was fitted separately). Please clarify. If all interaction terms were fitted, I would really worry about overfitting the model.

- Table 7: I had a hard time to understand and interpret this table. Can the authors present the log-likelihood ratio test where they compare the model without any interaction term, and the model with interaction terms with each of the sociodemographic variables?

Discussion:

I think some of the discussion sections need to be revisited once the authors clarify the above questions related to methodology and results.

- Last two second paragraph: Can the authors please elaborate the claim on “women with higher education levels have a greater understanding of the costs associated with delivery care”? I don’t think it’s a good idea to fit the interactions between costs and attributes. Moreover, the mixed effects results (Table 5) show that there is no significant preference heterogeneity for costs (as well as distance, medical equipment and supplies)- then why fit interaction terms? Also, do the stratified analyses by the major sociodemographic variables provide similar results? Can the authors please provide the stratified analysis results as the appendix for the readers to compare?

- Last paragraph: I am very unclear about the authors’ main point here. In the Result section, the authors describe that “female headed households had a statistically significant disutility for high delivery costs”… first of all, it is very unclear what does “positive” utility coefficient means for the households headed by males. Do the authors suggest that male headed households (which seem the majority) prefer to have a higher cost, which then relieve some of the cost-related burden? Please be more specific how the preference of partners or being in male-headed households may affect women’s financial decisions or concerns

Minor comments:

There are many grammatical errors and unclearly written sentences so I would recommend that this paper is to be carefully proofread.

- Please check the references throughout the paper. It looks like the paper used a citation manager program, and some references did not properly show up. For example, in the first paragraph, the citations “((3); (4); (5);” should be “(3-5”); {United Nations, 2014 #209} should be (X) with the correct reference number.

- 1st paragraph: “Rapid Urbanization” should be “Rapid urbanization”. Throughout the paper, there are many terms, which should not be capitalized when used in the middle of sentences.

- 2nd paragraph in Introduction: “One study reported a maternal mortality ratio of 700 for every 1,000 births” -> this seems like an error. Do the authors mean 700 per every 100,000 births?

- 2nd para.: “Kenyan Government abolished” -> “… removed” or “made delivery free of charge at public health…”

- 2nd para.: “However evidence”-> “However, evidence”

- Conditional model: the variable names are difficult to understand. Please define proper acronyms and refer to them.

- Results: please include percentages for those married and with secondary education in the text.

- Table 3: “Planning for delivery” seems confusing. Does that refer to the previous pregnancies women had or for future pregnancy? / please define all acronyms in the footnote / I recommend changing “Main earner status” to “Household head”, as that’s what the authors refer in the results and discussion. / Add “ago” after “Moved to the area over 5 years”

- Table 6: it should be labelled as from “women in the mixed logit model”, not “women in the conditional model”

6. PLOS authors have the option to publish the peer review history of their article (what does this mean?). If published, this will include your full peer review and any attached files.

Reviewer #1: No

---

## [Author Response · Author response to Decision Letter 0]

4 Mar 2020

RESPONSE TO REVIEWER#1

Review Comments to the Author

Reviewer #1: Thank you very much for the chance to review the paper by Ms. Aridi et al. on eliciting women’s preferences for a place of delivery in Kenya using a discrete choice experiment. Overall, the paper reports women’s preference and relative importance for a few key attributes and costs that women may consider when deciding a delivery place in a peri-urban setting in Kenya. The author presented interesting study design, analysis and results from a reasonably well-represented study population to address the authors’ research question. However, the paper has some methodological weaknesses, incomplete and unclear presentation of the data and analytical techniques, and questionable interpretation of the results, which significantly weaken the quality of the manuscript. There are a number of places where the sentences were poorly written, and lack of line numbers also made difficult to make comments.

MAJOR COMMENTS

Study Design

Comment#1

Although the authors indicated that they selected the attributes and attribute levels were selected based on a qualitative study, there are no description how a qualitative study was done. Please provide detailed information about the qualitative studies (the type of the qualitative study, how many interviews were done, brief sociodemographic characteristics of these women, and how these data were analyzed).

Response to comment#1

Thank you for your comment. A description of how the qualitative study was conducted has been added to the revised manuscript. The qualitative study was a phenomenological study and aimed to describe women’s experiences and perceptions of quality of care issues influencing their choice of a delivery facility. Six focus group discussions were done, one each, at 6 different facilities. These facilities represented the range of options available to women in this slum area. 40 women were purposively selected. The characteristics of the women interviewed as well as details on how the data was analyzed have been included in a draft manuscript attached as appendix 1. 

Comment #2

I strongly think the two attributes, ‘Interpersonal treatment at the health facility’ and ‘Attitude of healthcare workers’ are correlated. How do these two attributes differ? Furthermore, I worry that the dichotomy of “good” vs” bad” interpersonal treatment seems too simplistic, and can be unclear and very subjective to respondents. Also, I suspect that’s one of the reasons why the authors see a much lower coefficient for the interpersonal treatment.

Response to comment#2

Your point is well taken and we appreciate the opportunity to explain. 

In order to address your concerns around the attribute correlation we calculated the Pearson’s Products Moment (PPM) to assess for interattribute correlation. We found out that the Pearson’s correlation coefficient for the interattribute correlation between the attribute was between the attribute levels of QUALCLICGOOD (quality of clinical services good) and ATTITUDEKIND (attitude of healthcare worker kind) was 0.286. The correlation coefficient was 0.308 for the attribute levels of QUALCLICBAD (quality of clinical services good) and ATTITUDEUNKIND (unkindandunsupportive attitude of healthcare worker). Both were found to be insignificant at the 0.05 % level (See Appendix 2 with correlation matrix from Stata Output).

When women spoke about their experiences at the health facility the category we labelled ‘interpersonal treatment’ reflected two dimensions or aspects of care, (1) how the health facility handled them as clients/customers and (2) their perception of the clinical treatment they received during labor and delivery. This differs in important ways from the attribute labeled ‘attitude of the healthcare workers’ which was reflected descriptions of mistreatment/disrespect and abuse during delivery care. Disrespectful care is prevalent in low income settings where this study was conducted and there is extensive literature on the construct of disrespect/mistreatment of women during their delivery process. See (Bohren et al., 2014) ; (Oluoch-Aridi, Smith-Oka, Milan, & Dowd, 2018). 

In our work we sought to capture the women’s voices. The women respondents identified a salient distinction between disrespectful care labeled ‘attitude of health care workers’ and the construct labeled ‘interpersonal treatment’. Our priority was to hear and reflect the women’s voices in a way addressed the contextually relevant semantic meaning. These sentiments expressed in Swahili, the language we used to conduct the interviews, are distinct. In trying to express these two concepts in English- our choice of labels for the attributes insufficiently conveyed the distinction present in the Kiswahili. These two constructs are particularly important because are amenable to action by hospital management and policy makers. DCE’s are supposed to be focused on attributes that are policy amenable (Mangham, Hanson, & McPake, 2008).The attributes are present in the WHO framework and recognized in that they represent patient centered tenets of the experience of care (WHO, 2016). The attribute we labelled as interpersonal treatment in represented in the WHO framework as evidenced based 

 In saying this we acknowledge the important distinction between a lay persons understanding and articulation staff behavior as well as their limited understanding of what constitutes technical excellence. To improve understanding of the attribute and its meaning, we have relabeled the attribute on interpersonal treatment at the health facility to ‘quality of clinical services during delivery’ with an aim to convey this distinction. We hope it relays a more accurate presentation of the women’s voices in English. 

The distinction between our attribute labelled interpersonal treatment and the attribute labelled attitude of health workers in the manuscript has been labelled in other literature A similar study done in Zambia identifies attributes that are similar but distinct and important to policy makers namely (1) the likelihood that the hospital staff will examine the child properly, (2) staff attitudes (Hanson, McPake, Nakamba, & Archard, 2005).And the distinction is mathematically and we are sorry we didn’t label it well enough to identify the distinction in English 

The identification of attributes levels within DCE’s is often simplistic. This is done to reduce the process of complex cognitive especially for women with lower education for ease of cognition and for the respondents to make choices (Mangham et al., 2008).This is consistent with other published literature 

See,(Kruk, Paczkowski, Mbaruku, de Pinho, & Galea, 2009); (Kruk et al., 2010); (Larson et al., 2015))for similar types of attribute labels for DCE’s focused on place of delivery. 

Comment#3

I think the biggest question for the DCE design is the attribute level of costs. Given that the authors say that the there are no delivery fees at public health facilities, why didn’t the authors include no cost as the level option? Are there any other associated fees women need to pay when they deliver at public health facilities? The authors did not provide a data on how many of these women had previously delivered at public health facilities, compared to private health facilities. If most of women deliver at public health facilities, I worry that the hypothetical scenarios of paying the extra money might have not been correctly judged by the respondents as designed. Also, I wonder how the authors chose the cut-offs for the costs (3000, 5000, 8000)? Please help the readers understand whether these are reasonable range of costs to deliver at the facilities (especially at the public health facilities).

Response to comment#3

In Kenya, there are several auxiliary costs associated with “free” delivery care. The rates of these auxiliary costs are variable and facility dependent, but they are pervasive. These costs may be clearly articulated at the facility - such as bed costs or charges for drugs or it may be just an arbitrary tax for delivery related materials such as diapers or cotton wool etc. 

We did not include “no cost” as a level option because despite the free maternity whenever women go to delivery they are subjected to some cost to cover these expenses. That said, we do agree that the concept of auxiliary cost is not well articulated in the background section. The inception of “free maternity care” did substantially reduce out of pocket costs for women, but it did not eliminate them. We felt like giving the option of no cost would erroneously lead the women to make choices presuming they experienced no cost while in real life situations they are subjected to costs. 

The cut-off for the costs were chosen using information provided by the women about how much they had spent as delivery costs in the public and private health facilities and both before and after the removal of user fees at public health facilities. We have included a paragraph in the methods section that shows that the cost estimates for delivery came from the qualitative study. For a review on how the implementation of the free maternity sometimes resulted in people paying fees see (Tama et al., 2018). 

We do have data on the proportion of women who delivered at public health facilities compared to private health facilities and have included it in Table 5. 60% of the women delivered at public health facilities with 40% at private health facilities.( See Appendix 3 on Stata output on place of delivery) 

Comment#4

The authors mentioned that they chose a random sample out of a larger household survey. Which household survey is this (please cite it and provide a brief description)? Also, please provide more description of how this random selection was done – was there any stratification by geographic region or any other characteristics? Was the survey used one fixed same questionnaire for everyone? Please specify.

Response to comment#4

The household survey tool was a composite that was based on the validated surveys; Kenya Demographic Health Survey (KDHS) 2014 (Kenya National Bureau of Statistics et al., 2015) and the African Population and Health Research Centers APHRC slum survey (2012) (APHRC), 2012). We used the questions that have been and added the APHRC slum survey to extend our understanding around economics 

 A paragraph in the methods section has been included to explain the sources of the survey questionnaire. Two citations have been provided to clarify where we obtained our survey from. We have also attached a copy of our household survey as an appendix in the main study. An explanation of the selection of participants into the study is explained below;

Random selection for the main survey

Dandora is only 4 square km and has a population of 151, 046 according to the 2009 census. The random selection of women in the survey was done by mapping out the Dandora area and generating 200 points on the geographic positioning system (GPS) from google earth. This file has been attached as Appendix 4 for the reviewer’s perusal. 

After this a list of randomly generated numbers was created and matched to the 200 points in the map. Starting from a selected point the enumerators were instructed to approach the households identified using the random numbers on the list. The sample size calculation for the main survey was done using a formula based on the one used by the Slum Survey 2012. The following was the formula 

n= Z2 1-α/2 p (1-p)

e2

n= the required sample size of the individuals in the target population

p= the expected rate or prevalence of the key indicator to be estimated. We used 61% of live births in the five years preceding the survey were delivered in a health facility (KDHS 2014)

Deff design effect - we used 1.5 (following APHRC (2012) 

e= margin of error to be tolerated at 95 % level of confidence

Z = critical value for the standard normal distribution corresponding to a type 1 error rate of α

The household survey was administered to a total of 4640 randomly selected women. Then we used the criteria for the selection of the women for the DCE’s- this criteria was women who had delivered within one year at the time of the survey (October 2017). For the DCE a small proportion of roughly 10% of women who met the criteria of having delivered within one year and those who were aged between 18-49 years were selected to select the 481 women from the 4640 women. 

Comment#5

Please give a detailed assumption and calculation for how the sample size was calculated using the de Dekker-Grob’s formula. What coefficients did the authors assume initially? Did the authors calculate based on the conditional logit model or mixed logit model for their initial sample size calculation? At what percent of power? What was the minimum sample size required for the study?

Response to comment#5

We appreciate your attention to detail and we have amended the manuscript to correctly state that we chose the Johnson and Orme method to calculate sample size, as opposed to Dekker-Grob’s formula. e (Johnson & Orme, 2003)

We used the rule by Johnson and Orme to suggest the sample size required for main effects depends on the number of choice tasks (t), the number of alternatives (a) and the number of analysis cells (c). The formula considers main effects C is equal to the largest number of levels for any of the attributes when considering all two way interactions. We anticipated two way interactions so we multiplied the largest level size 3 by 2=6

We had 16 Choice tasks (t) with 3 alternatives (a) and 3*2 analysis cells (c). We then used the formula by Johnson and Orme to calculate the sample size, as shown below:

= N> 500*c/t*a

= N> 500*6/16*3

=N> 62.5

We also looked at the Johnson and Orme article and it recommended that sample sizes 500 and above were a minimum threshold that were sufficient to estimate effects, this sample size remains higher than other sample sizes used in estimating DCEs Louivere (2010) Bliemer and Rose (2009) Our sample size targeted sample size of 481 and eventual response rate of 421 (87.5%) was large enough power to provide results that were statistically significant for all relevant attributes. We did not use the debekker Grob in our original estimation for the conditional logit versus the mixed logit however as you see from our data the sample size was sufficient to allow us to estimate and interpret the conditional logit and mixed logit model

Comment #6

Can the authors provide more details about the D-efficient design? Were orthogonality and level of balance achieved? How many times did each attribute level appear in the questionnaire? What was the goodness-of-fit for the test?

Response to comment#6

Design balance =A balanced design has each attribute level occurring equally often across all pairs of attributes (Mangham 2009). Our design showed that almost half of the attribute levels occurred 16 times (equally often)

Orthogonality = the number of times that each attribute levels appears. Orthogonality is a desirable property of experimental designs that requires strictly independent variation of levels across attributes, in which each attribute level appears an equal number of times in combination with all other attribute levels

We achieved a reasonable level of orthogonality and balance, the D-efficiency level. Below we document 

• The alternative specific constant appears 16 times for the opt-out option and 16 times each for both the alternative A and B.

• For the attribute on the quality of clinical services during delivery- the attribute level good quality of clinical services during delivery appeared 19times while the attribute level bad quality of clinical services during delivery appeared 14 times. 

• For the attribute on the cleanliness of the health facility- the attribute level of clean appeared 16 times and the attribute level of dirty appeared 16 times. 

• For the attribute on kindness and supportive health worker the attribute on kind and supportive health worker appeared 16 times while unkind and unsupportive health worker appears 15 times 

• For the attribute on availability of medical equipment and drugs, available appeared 18 times and unavailability of medical equipment and drugs appeared 14 times 

• For the attribute on distance , short distance appeared 19 times and long distance appeared 13 times 

• For the attribute on the cost of delivery services 3000Ksh appeared 15 times ad 5000Ksh appeared ten times and 8000 appeared 7 times 

Attribute Attribute Level Coding Number of times attribute level appears in the experimental design

1. Opt-out Opt-out 0 16

 Alternative A 1 16

 Alternative B 1 16

2. Quality of clinical delivery services Good 1 19

 Bad 0 14

3. Cleanliness Clean 1 16

 Dirty 0 16

4. Attitude of health care worker Kind and supportive 1 16

 Unkind and unsupportive 0 15

5. Availability of medical equipment and drugs Available 1 18

 Unavailable 0 14

6. Distance to the health facility Short distance 1 19

 Long distance 0 13

7. Costs of delivery services (In Kenya shillings) 3000 15

 5000 10

 8000 7

Comment #7

- I am not sure why the authors would have blocked the questions into the two groups and ask each women answer two blocks anyway (it deficits the purpose of blocking…).

Response to comment #7

This is a semantic error and has been corrected to reflect the fact that women answered questions from one block each (8 questions each). This has been corrected to state, “…..The choice-sets were grouped into two, and each woman answered eight questions in a block…:”

Comment#8

- The authors indicated conducting a pilot study. Can the authors provide more details how the results from the pilot study testing were used to improve the study questionnaire?

Response to comment#8

The authors conducted a pilot study on a sample size of 30. In the pilot 6 attributes were selected for inclusion into the DCE. The pilot allowed us to text the most important and policy relevant attributes the coefficients that we found during the pilot were all significant even at the small sample size of 30, and hence we went ahead to conduct the experiment. 

Comment #9

The formula for possible choices is incorrect: it should be “2^5 x 3^1 = 96” for two scenarios matched 96*95/2= 4560 possible choices in the full factorial

Response to comment#9 

This formula has been corrected to reflect the correct number for the full fractional design. 

RESULTS SECTION 

Comment#1

The authors described that “opt-out” option was included in the model. How many people chose “opt-out” option out of all available choice tasks? Also, with about 26% of women planning not to deliver at health facilities, I worry whether there are other factors that influence women’s choices to deliver outside of health facilities (i.e. home). Did the authors look at preference coefficients among women who chose “opt-out” option at least once? Was there anyone who mostly chose the “opt-out” option?

Response to comment#1

Thank you for pointing this out, about one third of the women opted out of the DCE sequence.

On the 26% that you point above, the planned health facility was a variable constructed 

but a response to the question on the variable on planned delivery health facility =Is this the health facility where you originally planned to give birth, or did you have to change your plans?

Comment #2

- Can the authors clarify the difference between Table 6 and 7?

Response to Comment#2

For increased clarity, the data was re-analyzed in Stata 15 with an aim to improve clarity. The limitation of the JMP software (SAS) does not allow stepwise regression to more clearly analyze and interpret the mixed logit model The original table 6 and 7 have been replaced with new tables 5 and 6 that address the results of the generalized mixed logit model and the mixed logit model with the interactions. 

Comment#3

- I disagree with the approach that the authors put all possible combinations of different interactions between sociodemographic variables and attributes. Even so, I encourage not to show all the non-significant (especially if not driven by specific hypotheses) results in Table 6. 

Response to Comment#3

We appreciate your comment and agree that the model might lead to hence significant bias. We re-analyzed the data with the interaction terms separately in Stata 15. We present two tables table 5 showing the mixed logit without interactions and a new table 6 that shows the model with interactions (that have been modelled individually)We have focused on sociodemographic variables that are likely to influence place of delivery in the literature such as age, marital status, main earner status.

Comment #4

What is log worth? The authors’ present acronym of the attributes which were not previously defined in the manuscript thus it is difficult to understand 

Response to Coment#4

Log worth is language that is used by the software JMP to describe the magnitude of the coefficient of the parameter estimate. We have used STATA 15 software to re-analyze the data. 

Comment#5

Table 6. Please present meaningful acronyms (then define them in the footnote) or full description of the attributes.

Response to comment#5

We have revised the acronyms of the attributes to be uniform throughout the manuscript and used easily understood nomenclature. A legend has been included at the end of the table 6

Comment#6

P-value should be presented up to 2-3 digits in the table and throughout the paper.

Response to comment#6

P-Values have been presented to 3 digits, this has been revised throughout the paper.

Comment#7

- I am surprised that the cost coefficient in the conditional logit model is positive. More confusingly, while the cost coefficient is positive in the conditional logit (Table 4), the cost coefficient in the mixed logit model is very significantly negative. Why is that?

Response to comment#7

The generalized mixed logit provides information on preference heterogeneity in the sample. The negative sign of the cost attribute for the mean parameter shows that on average women had a utility for lower cost. The positive sign in the conditional model suggests that the women had a utility for high costs. We agree that this might present some confusion on women’s actual preferences. 

We attribute some of the weaknesses in the experimental design on the cost attribute might have demonstrated itself mathematically. Our experimental design that gave an unbalanced number of attribute levels for the attribute. We realized that the parameters might be biased and hence we will make our interpretations for the cost value cautiously. We have decided to focus on the other parameter estimates that were well specified. 

Additionally these signs may also represent the confusion by the low income women in this setting on the actual costs of delivery and how to value them. The true cost is a difficult issue because the marginal cost were hidden in real life until after the delivery. 

See quotation on confusion on costs from our qualitative study

Costs of delivery services. Overall costs were a major concern for women in both settings. This included direct costs related to financing the actual delivery service, but also indirect costs like transportation costs or opportunity costs associated with seeking delivery care. Sometimes additional costs were introduced inadvertently by some of the staff in public health facilities, leading to confusion and misunderstandings in health facilities that were advertised to be free. An example of a misunderstanding is illustrated below: 

"…To begin with, at the public hospital they told me that I had to pay 4,800 Ksh (48 USD) for the delivery service. My husband asked what it was for and they reduced it to 800 Ksh. He still insisted on knowing what the charges were for since he knew at a public health institution, maternity services were free of charge. In the end, he did not pay even a single cent…"

(FGD public tertiary health facility, rural setting)

With 20,208 observations we had sufficient power to discern the cost value when there is rational correspond in the real world. 

Comment#8

Table 6: It was not clear whether all interaction terms were fitted into one big model or whether each interaction was fitted separately). Please clarify. If all interaction terms were fitted, I would really worry about overfitting the model.

Response to comment#8

This has been addressed the authors have re-ran the mixed logit model again with interactions one at a time using Stepwise regression. We only focused on the sociodemographic attributes that have been proven to have an interaction with the attributes for place of delivery such as age, marital status main earner status. We have also only focused to show the significant and a few non-significant covariates for comparison. You can review this data in the new table 6. 

Comment#9

Table 7: I had a hard time to understand and interpret this table. Can the authors present the log-likelihood ratio test where they compare the model without any interaction term, and the model with interaction terms with each of the sociodemographic variables?

Response to comment#9

The likelihood ratio (LR) test is commonly used to evaluate the difference between nested models. One model is considered nested in another if the first model can be generated by imposing restrictions 

 On the parameters of the second. Most often the restriction is that the parameter is equal to zero. Hence does constraining the parameters to zero by leaving out the predictor variables significantly reduce the fit of the model. 

Hence we re-ran two models to assess the explanatory power for the models. 

The log-likelihood ratio for the generalized mixed logit model without any interactions was -2813.69 it was compared with the different log-likelihood ratios for the model with interactions. See Table 1 below.

The model’s explanatory power was improved for most of the interactions (see highlighted in Table 1 below) with the exception of the attributes related to attitude of health care worker. The LR test compares the log likelihoods of the two models and tests whether this difference is statistically significant 

Table 1. Log likelihood ratios for the mixed logit with interactions. 

Interaction terms Log Likelihood ratio for generalized mixed model with interactions

Conditional logit model -3396.87

Generalized mixed logit -2786.00

Clean_Age. -2845.93

Clean_Marr -3067.24

Clean_Main -3691.09

Medequip_Age -3861.89

Medequip_Marr -3835.91

Medequip_Main -3834.97

Att_Age -2801.22

Att_Marr -2786.00

Att_Main -2786.00

QualClin_Age -3834.90

QualClin_Marr -3837.18

QualClin_Main -3836.93

DISCUSSION SECTION 

I think some of the discussion sections need to be revisited once the authors clarify the above questions related to methodology and results.

The entire discussion section was revised to correspond to the re-analysis of the data. 

Comment#1

Last to second paragraph: Can the authors please elaborate the claim on “women with higher education levels have a greater understanding of the costs associated with delivery care”? I don’t think it’s a good idea to fit the interactions between costs and attributes. Moreover, the mixed effects results (Table 5) show that there is no significant preference heterogeneity for costs (as well as distance, medical equipment and supplies) then why fit interaction terms? Also, do the stratified analyses by the major sociodemographic variables provide similar results? 

Response to comment#1

We re-ran the data and elaborated different claims with regard to the attributes. Since women with higher education have been identified in other settings to have strong preference for better quality delivery services in DCE’s conducted in similar neighboring sub-Saharan African countries such as Tanzania and Ethiopia, we focused our analysis on the sociodemographic variables of age, marital status and main income earner status. (Kruk, Paczkowski, Mbaruku, de Pinho, & Galea, 2009); (Kruk et al., 2010)

Comment#2

Can the authors please provide the stratified analysis results as the appendix for the readers to compare?

Response to comment#2

An appendix with the stratified analysis for the major sociodemographic variables has been included as an appendix 5

Comment#3

Last paragraph: I am very unclear about the authors’ main point here. In the Result section, the authors describe that “female headed households had a statistically significant disutility for high delivery costs”… first of all, it is very unclear what does “positive” utility coefficient means for the households headed by males. Do the authors suggest that male headed households (which seem the majority) prefer to have a higher cost, which then relieve some of the cost-related burden? Please be more specific how the preference of partners or being in male-headed households may affect women’s financial decisions or concerns

Response to comment#3

We revised the earlier assertions that had been made based on the previous interpretations of the data

MINOR COMMENTS SECTION 

Minor comment#1

There are many grammatical errors and unclearly written sentences so I would recommend that this paper is to be carefully proofread.

Response to minor comment#1: 

The manuscript has been carefully proofread and the grammatical errors and unclearly written sentences have been revised

Minor Comment#2

- Please check the references throughout the paper. It looks like the paper used a citation manager program, and some references did not properly show up. For example, in the first paragraph, the citations “((3); (4); (5);” should be “(3-5”); (UnitedNations, 2014) should be (X) with the correct reference number.

Response to minor comment#2

• The citations for 3-5 have been edited to be “(3-5)” as recommended 

• The United Nations reference for 2014 has been revised to a numerical

Minor Comment#3

- 1st paragraph: “Rapid Urbanization” should be “Rapid urbanization”. Throughout the paper, there are many terms, which should not be capitalized when used in the middle of sentences.

Response to minor comment#3 

• Rapid Urbanization has been changed to Rapid urbanization. 

• The capitalizations in the middle of sentences have been revised

Minor comments#4

- 2nd paragraph in Introduction: “One study reported a maternal mortality ratio of 700 for every 1,000 births” -> this seems like an error. Do the authors mean 700 per every 100,000 births?

- 2nd para: “Kenyan Government abolished” -> “… removed” or “made delivery free of charge at public health…”

- 2nd Para. “However evidence”-> “However, evidence”

Response to minor comments#4 

• The maternal mortality ratio has been adjusted 700 per every 100,000 births. 

• The phrase Kenyan Government abolished has been changed to “ made delivery free of charge at public health facilities”

• In the phrase However evidence a comma has been inserted and this has been changed to However, evidence 

Minor Comment #5 

- Conditional model: the variable names are difficult to understand. Please define proper acronyms and refer to them.

Response to minor comment#5

- The variable names have been changed to proper acronyms for example Cleanliness is now clean. A legend has been created underneath the tables to refer to the acronyms that have been used. 

RESULTS 

Minor comment#6

• Please include percentages for those married and with secondary education in the text.

Response to minor comment#6

• The percentages for those married and with secondary education in the text as requested. 

Minor Comment#7 

• Table 3: “Planning for delivery” seems confusing. Does that refer to the previous pregnancies women had or for future pregnancy? 

Response to minor comment#7

• The term “ planning for delivery” has been revised to refer to previous pregnancies 

Minor comment#8

• Please define all acronyms in the footnote.

Response to minor comment #8

• All the acronyms have been included in the footnote. 

Minor comment#9

• I recommend changing “Main earner status” to “Household head”, as that’s what the authors refer in the results and discussion. / Add “ago” after “Moved to the area over 5 years”

Response to minor comment#9

• “Main earner status” was a different variable from the “household head”. The main earner status assessed whether the woman is the main earner in the household> The household head assessed 

• The word “ ago” has been added to the phrase “ Moved to the area over 5 years” 

Minor Comment#10 

• Table 6: it should be labelled as from “women in the mixed logit model”, not “women in the conditional model”

Response to minor comment#10

• The label for Table 6 has been changed to the “ women in the mixed logit model” 

REFERENCES

(APHRC), (2012). Population and Health Dynamics in Nairobi's Informal Settlements Report of the Nairobi Cross-sectional Slums Survey(NCSS)2012. 

Bohren, M. A., Hunter, E. C., Munthe-Kaas, H. M., Souza, J. P., Vogel, J. P., & Gulmezoglu, A. M. (2014). Facilitators and barriers to facility-based delivery in low- and middle-income countries: a qualitative evidence synthesis. Reprod Health, 11(1), 71. doi:10.1186/1742-4755-11-71

Hanson, K., McPake, B., Nakamba, P., & Archard, L. (2005). Preferences for hospital quality in Zambia: results from a discrete choice experiment. Health Econ, 14(7), 687-701. doi:10.1002/hec.959

Kenya National Bureau of Statistics, Ministry of Health/Kenya, National AIDS Control Council/Kenya, Kenya Medical Research Institute, Population, N. C. f., & Development/Kenya. (2015). Kenya Demographic and Health Survey 2014. Retrieved from Rockville, MD, USA: http://dhsprogram.com/pubs/pdf/FR308/FR308.pdf

Kruk, M. E., Paczkowski, M., Mbaruku, G., de Pinho, H., & Galea, S. (2009). Women's preferences for place of delivery in rural Tanzania: a population-based discrete choice experiment. Am J Public Health, 99(9), 1666-1672. doi:10.2105/ajph.2008.146209

Kruk, M. E., Paczkowski, M. M., Tegegn, A., Tessema, F., Hadley, C., Asefa, M., & Galea, S. (2010). Women's preferences for obstetric care in rural Ethiopia: a population-based discrete choice experiment in a region with low rates of facility delivery. J Epidemiol Community Health, 64(11), 984-988. doi:10.1136/jech.2009.087973

Larson, E., Vail, D., Mbaruku, G. M., Kimweri, A., Freedman, L. P., & Kruk, M. E. (2015). Moving Toward Patient-Centered Care in Africa: A Discrete Choice Experiment of Preferences for Delivery Care among 3,003 Tanzanian Women. PloS one, 10(8), e0135621. doi:10.1371/journal.pone.0135621

Mangham, L. J., Hanson, K., & McPake, B. (2008). How to do (or not to do) … Designing a discrete choice experiment for application in a low-income country. Health policy and planning, 24(2), 151-158. doi:10.1093/heapol/czn047

Oluoch-Aridi, J., Smith-Oka, V., Milan, E., & Dowd, R. (2018). Exploring mistreatment of women during childbirth in a peri-urban setting in Kenya: experiences and perceptions of women and healthcare providers. Reprod Health, 15(1), 209. doi:10.1186/s12978-018-0643-z

Tama, E., Molyneux, S., Waweru, E., Tsofa, B., Chuma, J., & Barasa, E. (2018). Examining the Implementation of the Free Maternity Services Policy in Kenya: A Mixed Methods Process Evaluation. International Journal of Health Policy and Management, 7(7), 603-613. doi:10.15171/ijhpm.2017.135

UnitedNations. (2014). Population Facts. Retrieved from New YorkAu: 

WHO. (2016). Standards for improving quality of maternal and newborn care in health facilities. Retrieved from

Johnson R, Orme B. Getting the most from CBC. Sequim: Sawtooth Software Research Paper Series, Sawtooth Software; 2003. [Google Scholar] 

Louviere, JJ., Hensher, D.A., SwaitJ.D., 2010 Stated Choice Mehtods: Analysis and Applications. Cambridge University Press.

Bliemer MCJ, Rose JM. Construction of experimental designs for mixed logit models allowing for correlation across choice observations. Transp Res B Methodol. 2010;44(6):720–734. doi: 10.1016/j.trb.2009.12.004

---

## [Decision Letter · Decision Letter 1]

16 Sep 2020

PONE-D-19-31801R1

Eliciting women’s preferences for place of delivery in a peri-urban setting in Kenya: A discrete choice experiment

PLOS ONE

Dear Dr. Aridi,

Thank you for submitting your manuscript to PLOS ONE. After careful consideration, we feel that it has merit but does not fully meet PLOS ONE’s publication criteria as it currently stands. Therefore, we invite you to submit a revised version of the manuscript that addresses the points raised during the review process.

The reviewer who provided the report during the first round of peer-review unfortunately was not available to re-assess the manuscript . However, your manuscripts has now been evaluated by four additional reviewers who provided positive comments regarding your revisions (their comments are available below). They also raised some concerns about methodological aspects of your study, especially the statistical analysis.  

Could you please revise the manuscript to carefully address the concerns raised?

We look forward to receiving your revised manuscript.

Kind regards,

Dario Ummarino, Ph.D.

Associate Editor

PLOS ONE

Reviewers' comments:

Reviewer's Responses to Questions

**Comments to the Author**

1. If the authors have adequately addressed your comments raised in a previous round of review and you feel that this manuscript is now acceptable for publication, you may indicate that here to bypass the “Comments to the Author” section, enter your conflict of interest statement in the “Confidential to Editor” section, and submit your "Accept" recommendation.

Reviewer #2: All comments have been addressed

Reviewer #3: All comments have been addressed

Reviewer #4: (No Response)

Reviewer #5: (No Response)

2. Is the manuscript technically sound, and do the data support the conclusions?

Reviewer #2: Yes

Reviewer #3: Yes

Reviewer #4: Yes

Reviewer #5: Partly

3. Has the statistical analysis been performed appropriately and rigorously? 

Reviewer #2: I Don't Know

Reviewer #3: Yes

Reviewer #4: Yes

Reviewer #5: Yes

4. Have the authors made all data underlying the findings in their manuscript fully available?

Reviewer #2: Yes

Reviewer #3: Yes

Reviewer #4: Yes

Reviewer #5: Yes

5. Is the manuscript presented in an intelligible fashion and written in standard English?

Reviewer #2: No

Reviewer #3: Yes

Reviewer #4: Yes

Reviewer #5: No

6. Review Comments to the Author

Reviewer #2: 1. The manuscript did not follow the journal guideline, therefore requesting the authors to revisit the guideline and improve it.

2. The in-text citation needs reworking, the brackets used are the same as the brackets used for none citation area for example abbreviations used () and in-text citation used (); authors could differentiate these by using [] in in-text citation. Also several citations used for one information to be under one bracket and not each on its own bracket.

3. In the experiment it was only one attribute which was objective others were subjective. Other attributes were very subjective; dirtiness, availability of equipment, distance. It is not mentioned on whether they were defined to the respondents or not.

4. I happened to review this paper submitted BMJ with manuscript number bmjopen-2020-038865, can authors explain why double submission?

Reviewer #3: The authors has addressed all of these comments. The manuscript has improved significantly. I have no further comments.

Reviewer #4: Thank you very much for reviewing this manuscript. This article describes preferences of Kenyan women when making choices about the place of birth. It is an interesting and relevant topic and the results give insight in important attributes for women when choosing a place of birth. This knowledge is important to provide woman-centred care and to improve maternal and newborn outcomes.

The previous reviewer assessed the manuscript very accurately and I think the researchers addressed these comments properly. These changes and additions contribute positively to the readability of the manuscript.

However, after reading this manuscript I have some additional comments and questions:

1. I would like to discuss the term ‘place of delivery’. To my opinion, women do not deliver a baby, but give birth to a baby. So, I would recommend to use the term ‘place of birth’ instead of ‘place of delivery’. I think this is more consistent with international literature in this area.

2. Line 157: the word ‘right’ is in the sentence twice.

3. Participants’ characteristics: this paragraph is not well structured and is difficult to understand. It seems that the percentages presented in line 234 and 235 belong to the study population who was selected for the interview (n=481). However, table 2 presents the same percentages and the number of participants in table 2 is 421. This is unclear and confusing, especially because the number of women who completed the DCE-questionnaire is 411. These women are suitable for the DCE-analysis.

I think you have to present the sociodemographic results of the total study population who completed the DCE-questions (n=411) in table 2 and 3.

Do you have any information about the women who declined participation (n=60) or who did not complete the DCE-questionnaire (N=10).

4. In table 2: The numbers presented at parity =1 and parity >2 count to 481. I think this is not correct. These numbers have to count to 411.

5. Table 2: Parity > 2 has to be parity ≥ 2 (gave birth 2 times or more)

Reviewer #5: This manuscript investigates Women’s preferences for place of delivery in a peri-urban setting in Kenya using a discrete choice experiment. I have below comments and questions.

Please ask an English editor to edit the writing.

Line 133, Appendix 2 does not have contents for design. But Appendix 2 shows significant correlations (p<0.0001) between a couple of attributes. Please discuss if their correlations would affect the model fitting and results.

Line 136, “(2^5 x 1^3) =96 = (96*95)/2 = 4560”, how can they equal with each other? 1^3 should be 3^1. They should be described clearly by separating the two parts of calculations.

Line 136, There are only three choices for delivery places. To avoid confusion, ”the number of possible choices” may be said as “the number of alternatives of attribute levels”.

Line 137, where dose 35 come from?

Line 142, how do you group the choice-sets? What eight questions in each group?

In the results, for the statement such as “statistically significant/insignificant at the 95% level” on page 14, do you mean the significance level was set as 95%? If the type I error was set at 0.05, the significance level should be stated as 5%.

Line 405-414, It is hard to find the matched supporting information.

7. PLOS authors have the option to publish the peer review history of their article (what does this mean?). If published, this will include your full peer review and any attached files.

Reviewer #2: No

Reviewer #3: No

Reviewer #4: No

Reviewer #5: No

---

## [Author Response · Author response to Decision Letter 1]

8 Oct 2020

Review Comments to the Author

Reviewer #2: 

Comment

1.The manuscript did not follow the journal guideline, therefore requesting the authors to revisit the guideline and improve it.

Response: The authors have reviewed the journal guidelines and followed them in an attempt to improve the manuscript. The abstract has been divided in the four sections of objective, methods, results and conclusion. The abstract word count of 300 words has been adhered to. All other areas that did not follow journal guidelines have been adjusted appropriately.

2.The in-text citation needs reworking, the brackets used are the same as the brackets used for none citation area for example abbreviations used () and in-text citation used (); authors could differentiate these by using [] in in-text citation. Also, several citations used for one information to be under one bracket and not each on its own bracket.

Response: The in-text citations have been reworked and the brackets used for the in-text citations have been distinguished between the other brackets used for non-citation. Several citations have been put under one bracket as advised.

3. In the experiment it was only one attribute which was objective others were subjective. Other attributes were very subjective; dirtiness, availability of equipment, distance. It is not mentioned on whether they were defined to the respondents or not.

Response: We appreciate the comment. Only one attribute was objective. We have included a sentence that illustrates how the subjective attributes were defined. These definitions were presented to the women on the choice-card. Women had the definitions prior to making their choices and this is stated in lines (157-159) See below

The attributes of the health facility were explained to the women using a choice-card that contained a brief description of the definition of the attributes. For example. Cleanliness meant a health facility that had a clean ward with clean beds, bathrooms and toilets (See Appendix 3). We have included an appendix 3 that shows the choice card that was shown to the women and how the subjective attributes were defined to the women to have a shared understanding.

4. I happened to review this paper submitted BMJ with manuscript number bmjopen-2020-038865, can authors explain why double submission?

Response: The study was part of a two-site study the BMJ manuscript presents the results of the Discrete Choice Experiment in Naivasha sub-County (named rural sub-County). The current manuscript reports on the results of the Discrete Choice Experiment within a peri-urban setting Embakasi-North sub County. We previously attempted to submit the results as a comparative study but was rejected and advised to handle the two sites separately. This is primarily because the attributes that were identified were different in each site and hence deemed incomparable directly. All five attributes were common for both sites with the exception of one attribute- cleanliness of the health facility in the peri-urban site and the availability of referral services in rural sub-County. 

Reviewer #3 

The authors have addressed all of these comments. The manuscript has improved significantly. I have no further comments.

Reviewer #4 

Thank you very much for reviewing this manuscript. This article describes preferences of Kenyan women when making choices about the place of birth. It is an interesting and relevant topic and the results give insight in important attributes for women when choosing a place of birth. This knowledge is important to provide woman-centered care and to improve maternal and newborn outcomes. The previous reviewer assessed the manuscript very accurately and I think the researchers addressed these comments properly. These changes and additions contribute positively to the readability of the manuscript.

However, after reading this manuscript I have some additional comments and questions:

1. I would like to discuss the term ‘place of delivery’. To my opinion, women do not deliver a baby, but give birth to a baby. So, I would recommend to use the term ‘place of birth’ instead of ‘place of delivery’. I think this is more consistent with international literature in this area.

Response: Thank you for your comments and your recommendation for consistency with international literature. We will change the term from “place of delivery to ‘place of child birth’

2. Line 157: the word ‘right’ is in the sentence twice.

Response: The word right has been deleted and only one ‘right’ is maintained

3. Participants’ characteristics: this paragraph is not well structured and is difficult to understand. It seems that the percentages presented in line 234 and 235 belong to the study population who was selected for the interview (n=481). 

However, table 2 presents the same percentages and the number of participants in table 2 is 421. This is unclear and confusing, especially because the number of women who completed the DCE-questionnaire is 411. These women are suitable for the DCE-analysis.

I think you have to present the sociodemographic results of the total study population who completed the DCE-questions (n=411) in table 2 and 3.

Do you have any information about the women who declined participation (n=60) or who did not complete the DCE-questionnaire (N=10).

Response: 

The paragraph on participants characteristics has been revised for clarity. We have revised the participant characteristics to be reflective of the sub-sample that actually completed the DCE N=411.

We conducted additional analysis on the women who declined participation n=60 and found that they had no significant differences between them and the participants in the study. (N=60)

There was also no difference between the women who completed and those that did not complete the DCE questionnaire. (N=10)

The women were dropped automatically for having dominant choices. 

4. In table 2: The numbers presented at parity =1 and parity >2 count to 481. I think this is not correct. These numbers have to count to 411.

Response: This comment is appropriate. The analysis has been revised to have the parity count = 411 taking into account only those women who completed the DCE questions 

5. Table 2: Parity > 2 has to be parity ≥ 2 (gave birth 2 times or more)

Response: The parity variable is Table 2 has been revised to be parity ≥ 2 (gave birth 2 times or more)

Reviewer #5

This manuscript investigates women’s preferences for place of delivery in a peri-urban setting in Kenya using a discrete choice experiment. I have below comments and questions.

Please ask an English editor to edit the writing.

Response: An English editor has edited the writing as requested. 

Line 133, Appendix 2 does not have contents for design. But Appendix 2 shows significant correlations (p<0.0001) between a couple of attributes. Please discuss if their correlations would affect the model fitting and results.

Response: The results were reviewed and we found that the correlations do not affect the model fitting or the results 

Line 136, “(2^5 x 1^3) =96 = (96*95)/2 = 4560”, how can they equal with each other? 1^3 should be 3^1. They should be described clearly by separating the two parts of calculations.

Response: The two parts of the calculation was separated as follows 

“(2^5 x 3^1) =96. The two alternatives choice-sets were calculated as follows (96*95)/2 = 4560”,

Line 136, There are only three choices for delivery places. To avoid confusion, ”the number of possible choices” may be said as “the number of alternatives of attribute levels”.

Response: the number of possible choices has been revised to “the number of alternatives of attribute levels”.

Line 137, where dose 35 come from?

Response: The 35 was an error and has been replaced with 16 which comes from the JMP software results that automatically calculated the results. 

Line 142, how do you group the choice-sets? What eight questions in each group?

Response: This is randomly determined by the software that we use open data kit (ODK). The sentence has been revised to include this detail. The choice-sets were grouped into two through a process called blocking. This is a standard process that is used to reduce the burden of answering many questions for each woman using the ODK software.

In the results, for the statement such as “statistically significant/insignificant at the 95% level” on page 14, do you mean the significance level was set as 95%? If the type I error was set at 0.05, the significance level should be stated as 5%.

Response: Yes, we meant that the significance was set at 95% and the significance is at the 5% level. This wording has been changed to reflect significance is at the 5%level.

For the generalized mixed multinomial logit model with no interactions, all the mean coefficients values for all the attributes, including the opt-out, were statistically significant at the 5% level

Line 405-414, It is hard to find the matched supporting information.

Response: An attempt has been made to match the supporting information and all the supporting information is included in the Appendices as listed.

---

## [Decision Letter · Decision Letter 2]

28 Oct 2020

Eliciting women’s preferences for place of childbirth in a peri-urban setting in Nairobi,  Kenya: A discrete choice experiment

PONE-D-19-31801R2

Dear Dr. Aridi,

We’re pleased to inform you that your manuscript has been judged scientifically suitable for publication and will be formally accepted for publication once it meets all outstanding technical requirements.

Kind regards,

Tanya Doherty, PhD

Academic Editor

PLOS ONE

Additional Editor Comments (optional):

Please address these final comments from  reviewer 5 below:

Page 22, the list of supporting information does not match the Appendix number. For example, below supporting information can not be found from this manuscript,

S1 Appendix. The Characteristics of women interviewed in the Focus Group Discussions

S4 Appendix. Sampling for the main household survey.

S5 Appendix. Dataset for the DCE and household survey for women in the peri-urban setting.

S7Additional analysis for the Mixed Logit model with interactions

S7 Appendix. Ethical approval form.

S8 Appendix. Informed consent form.

Appendix S2 should Appendix 7. The DCE Experimental design

Appendix S9 should S10. DCE Choice card Information packet.

Appendix S10 should S9. Published qualitative paper.

Reviewers' comments:

Reviewer's Responses to Questions

**Comments to the Author**

1. If the authors have adequately addressed your comments raised in a previous round of review and you feel that this manuscript is now acceptable for publication, you may indicate that here to bypass the “Comments to the Author” section, enter your conflict of interest statement in the “Confidential to Editor” section, and submit your "Accept" recommendation.

Reviewer #5: (No Response)

2. Is the manuscript technically sound, and do the data support the conclusions?

Reviewer #5: (No Response)

3. Has the statistical analysis been performed appropriately and rigorously? 

Reviewer #5: (No Response)

4. Have the authors made all data underlying the findings in their manuscript fully available?

Reviewer #5: (No Response)

5. Is the manuscript presented in an intelligible fashion and written in standard English?

Reviewer #5: (No Response)

6. Review Comments to the Author

Reviewer #5: Page 22, the list of supporting information does not match the Appendix number. For example, below supporting information can not be found from this manuscript,

S1 Appendix. The Characteristics of women interviewed in the Focus Group Discussions

S4 Appendix. Sampling for the main household survey.

S5 Appendix. Dataset for the DCE and household survey for women in the peri-urban setting.

S7Additional analysis for the Mixed Logit model with interactions

S7 Appendix. Ethical approval form.

S8 Appendix. Informed consent form.

Appendix S2 should Appendix 7. The DCE Experimental design

Appendix S9 should S10. DCE Choice card Information packet.

Appendix S10 should S9. Published qualitative paper.

7. PLOS authors have the option to publish the peer review history of their article (what does this mean?). If published, this will include your full peer review and any attached files.

Reviewer #5: No

---

## [Editor Report · Acceptance letter]

25 Nov 2020

PONE-D-19-31801R2 

Eliciting women’s preferences for place of child birth at a peri-urban setting in Nairobi, Kenya: A discrete choice experiment 

Dear Dr. Oluoch-Aridi:

I'm pleased to inform you that your manuscript has been deemed suitable for publication in PLOS ONE. Congratulations! Your manuscript is now with our production department. 

Kind regards, 

on behalf of

Professor Tanya Doherty 

Academic Editor

PLOS ONE